# Robust Quantum Search with Uncertain Number of Target States

**DOI:** 10.3390/e23121649

**Published:** 2021-12-08

**Authors:** Yuanye Zhu, Zeguo Wang, Bao Yan, Shijie Wei

**Affiliations:** 1State Key Laboratory of Low-Dimensional Quantum Physics and Department of Physics, Tsinghua University, Beijing 100084, China; zhuyy16@mails.tsinghua.edu.cn (Y.Z.); wzg17@mails.tsinghua.edu.cn (Z.W.); 213090525@seu.edu.cn (B.Y.); 2State Key Laboratory of Mathematical Engineering and Advanced Computing, Zhengzhou 450001, China; 3Beijing Academy of Quantum Information Sciences, Beijing 100193, China

**Keywords:** quantum algorithm, quantum computation, quantum information

## Abstract

The quantum search algorithm is one of the milestones of quantum algorithms. Compared with classical algorithms, it shows quadratic speed-up when searching marked states in an unsorted database. However, the success rates of quantum search algorithms are sensitive to the number of marked states. In this paper, we study the relation between the success rate and the number of iterations in a quantum search algorithm of given $\lambda =\sqrt{M/N}$, where *M* is the number of marked state and *N* is the number of items in the dataset. We develop a robust quantum search algorithm based on Grover–Long algorithm with some uncertainty in the number of marked states. The proposed algorithm has the same query complexity $O\left(\sqrt{N}\right)$ as the Grover’s algorithm, and shows high tolerance of the uncertainty in the ratio M/N. In particular, for a database with an uncertainty in the ratio $\frac{M\pm \sqrt{M}}{N}$, our algorithm will find the target states with a success rate no less than 96%.

## 1. Introduction

The quantum search algorithm is one of the most significant quantum algorithms [1]. Compared with classical search algorithms, quantum search algorithms exhibit quadratic speedup [2,3]. This demonstrates the superiority of quantum computing over classical computing. Grover proposed the first quantum search algorithm [1,4], which can find *M* marked items from an unstructured database with *N* items by querying only $O(\sqrt{N/M})$ times [5,6]. If the measurements are made after the optimal iterations, Grover’s algorithm will have a success rate $P_{max}=\sin^{2}\left[\left(2j_{op}+1\right)\beta \right]$ to find the marked items, where $\beta =\arcsin\sqrt{M/N}$ and $j_{op}=\left[\left(\pi /2-\beta \right)/\left(2\beta \right)\right]$ is the number of optimal Grover iterations. If $(2j_{op}+1)\beta \approx \pi /2$, the maximum probability approaches 1, which means that the Grover’s algorithm usually has a high success rate if the dimension of the quantum database is very large.

There have been several important developments in the Grover’s algorithm. In some situations, such as structured search [7], where the success rate is the product of the success rates of individual search, high success rate in each individual search is critical; especially, when dimensions are not so large, the standard Grover’s algorithm will not perform well. In order to solve this problem, some modified search algorithms have been proposed [8,9,10,11,12]. The Grover–Long algorithm [11], one of these improved algorithms, has been proved to be the simplest and most optimal [13,14]. This algorithm achieves 100% success rate, whereas Grover’s algorithm can only achieve 100% success rate when finding one out of four.

In both the original Grover and improved versions of Grover’s algorithm, one needs to know the exact number of marked states in advance. Therefore, if the exact number is not known, these algorithms can not determine when to stop [15]. Spatial search [16,17,18] is one of the methods to solve this problem. Fixed-point search algorithm is another method to solve this problem. By constructing the recursively searching operator, the ratio of marked state always amplifies after each search, the π/3 fixed-point search algorithm of Grover [19], for example. In this algorithm, each search approaches the marked states monotonously, but the cost of monotony is large in this algorithm, and the quadratic speedup of standard Grover’s algorithm is lost.

The Yoder–Low–Chung algorithm [20] was proposed to improve the performance of fixed-point algorithms on wide ranges of M/N. It retains the quadratic speedup advantage of quantum search, and it achieves the fixed-point property at the same time. It also solves overcooking problem, but the success rate of which is not monotonically increasing as in the π/3 algorithm. The error is bounded by a tunable parameter δ∈[0,1] over an ever-widening range of M/N, but the phases in each search step need to be calculated by solving a hype-trigonometric equation.

In this paper, we develop a robust quantum search algorithm, based on the Grover–Long algorithm, which overcomes the problem of not knowing the exact ratio M/N in advance. This algorithm has the advantage of easiness in constructing the search operators, and also certain degrees of “fixed-point” properties. Namely, it enjoys a high success rate over a wide range of the ratio of M/N. In our algorithm, we do not need to know the exact number of marked states, but rather an approximate number in the range $\left[\lambda_{0},\lambda_{0}+\Delta \right]$ of the ratio λ=M/N. The error of our algorithm is bounded by a parameter δ∈[0,1] related to the $\Delta /\lambda_{0}$. Specifically, searching operators in our algorithm are determined by the lower bound $\lambda_{0}$. After $J+1-J_{D}$ iterations, the probability of success is larger than $1-\delta^{2}$, where $J_{D}$ is denoted as $\left\lceil \left(\frac{1}{2\sqrt{\lambda_{0}+\Delta }}+\frac{4}{\pi }\right)\delta \right\rceil$.

This paper is organized as follows. First of all, the Grover–Long algorithm is summarized. Secondly, the relationship between the success rate of Grover–Long algorithm and iterative steps is studied, and the relation between iteration number and success rate is given. Thirdly, we propose a robust search version of Grover–Long algorithm and show its high tolerance to the ratio M/N. The comparison is then made with the Yoder–Low–Chuang algorithm, standard Grover’s algorithm, and the Grover π/3 fixed-point algorithm, respectively. Finally, we prove that our algorithm can find the target state with a success rate of more than 96% from an database, with only an estimate of *M*, in the range between $\left\{M-\sqrt{M},M+\sqrt{M}\right\}$, which can be carried out using the quantum counting algorithm [21,22].

## 2. Overview of Grover–Long Algorithm

The Grover–Long algorithm can extract *M* marked items from an unstructured database with *N* items by querying $O\left(\sqrt{N/M}\right)$ times. First, from the given ratio M/N, parameter β can be calculated:(1)$$\beta =\arcsin\sqrt{\frac{M}{N}},$$
which is further used to determine the value of search steps $j_{op}=\left\lfloor \frac{\pi -2\beta }{4\beta }\right\rfloor$. The square brackets here represent the floor function. One can set the number of iteration as
(2)$$J\geq j_{op}.$$Then, the phases in the search algorithm are calculated by
(3)$$\phi =2\arcsin\left(\frac{\sin\frac{\pi }{4J+6}}{\sin\beta }\right).$$Next, the oracle operator can be expressed as
(4)$$I_{\tau }=I+\left(e^{i\phi }-1\right)|\tau \rangle \langle \tau |,$$
where |τ〉 denotes as the superposition of *M* marked states, and the phase shifting operator for the |0〉 state is
(5)$$I_{0}=I+\left(e^{i\phi }-1\right)|0\rangle \langle 0|.$$Finally, the Grover–Long operator in each iteration is
(6)$$Q=-HI_{0}HI_{\tau },$$
where *H* is the Hadamard gate. After J+1 steps of iterations, one can obtain the marked states with certainty by measurement. When ϕ=π, we recover the original Grover’s algorithm, which usually does not find the marked states with certainly.

The quantum search algorithm can be described using the SO(3) picture [11,23] instead of SU(2). In this picture, the quantum search operator in Equation (Equation 6) corresponds to a rotation in three-dimensional space with the following matrix form
(7)$$R_{Q}=\left[\begin{array}{ccc} R_{11} & R_{12} & R_{13}\\ R_{21} & R_{22} & R_{23}\\ R_{31} & R_{32} & R_{33} \end{array}\right],$$
where the entries of the matrix $R_{Q}$ are calculated in [11]. The states are rotated along the $\vec{l}$ axis
(8)$$\vec{l}=\left[\begin{array}{c} \cos\frac{\phi }{2}\\ \sin\frac{\phi }{2}\\ \cos\frac{\phi }{2}\tan\beta \end{array}\right],$$
with an angle
(9)$$\alpha =4\arcsin\left[\sin\left(\frac{\phi }{2}\right)\sin\beta \right]=\frac{2\pi }{2J+3}.$$In this picture, the state vector $|\psi \rangle =\left(a+bi\right)|\tau \rangle +\left(c+di\right)|\bar{\tau }\rangle$ is represented as
(10)$${\vec{r}}_{\psi }=\langle \psi |\vec{\sigma }|\psi \rangle =\left[\begin{array}{c} 2(ac+bd)\\ 2(-bc+ad)\\ a^{2}+b^{2}-c^{2}-d^{2} \end{array}\right],$$
where $\vec{\sigma }=\sigma_{x}\vec{i}+\sigma_{y}\vec{j}+\sigma_{z}\vec{k}$ and $\vec{i}$, $\vec{j}$, $\vec{k}$ are the unit vector along the *x*, *y*, *z* axis. The initial state $|\psi_{i}\rangle$ and the marked state |τ〉 are represented by
(11)$${\vec{r}}_{i}=\left[\begin{array}{c} \sin(2\beta )\\ 0\\ -\cos(2\beta ) \end{array}\right],\;{\vec{r}}_{f}=\left[\begin{array}{c} 0\\ 0\\ 1 \end{array}\right].$$Each search step is a rotation of ${\vec{r}}_{\psi }$ toward ${\vec{r}}_{f}$. The SO(3) description of the Grover–Long algorithm is pictured as a circle in Figure 1.

## 3. Relationship between the Success Rate and Searching Iterations

During each search step of the Grover–Long algorithm, state ${\vec{r}}_{\psi }=\bar{OA}$ rotates toward ${\vec{r}}_{f}=\bar{OB}$, which is described geometrically in Figure 1. In other words, this process is point *A* moving to point *B* on the blue circle $⊙r_{0}$. In this picture, the probability of finding the marked state is $(z_{A}+1)/2$ [11], where $z_{A}$ is the *Z* component of point *A*. Thus, if one wants to find a marked state with a probability greater than $1-\delta^{2}$, point *A* of the segment $\bar{r_{o}A}$ must rotate into the arc $\overset{⌢}{CD}$, where point *C* and point *D* are the intersections of the circle $⊙r_{o}$ and the red error circle: $x^{2}+y^{2}+{\left(1-2\delta^{2}\right)}^{2}=1$. If we can calculate the arc length $\overset{⌢}{CD}$, then we obtain a reasonable number of iterations.

For this reason, we focus on the spherical cone, which consists of the unit sphere and the red circle. It is shown in Figure 2. The segment $\bar{BC}$ in Figure 2 is a segment from point B(0,0,1) to point *C* on the red error circle. In Figure 2, $\bar{OD}=1-2\delta^{2}$, $\bar{OC}=1$, $\bar{BD}=2\delta^{2}$. The following relationship holds:(12)$$\begin{array}{cc} \bar{CD} & =\sqrt{{\bar{OC}}^{2}-{\bar{OD}}^{2}}\\ & =\sqrt{1-{\left(1-2\delta^{2}\right)}^{2}}=\sqrt{4\delta^{2}-4\delta^{4}}, \end{array}$$
(13)$$\begin{array}{cc} \bar{BC} & =\sqrt{{\bar{CD}}^{2}+{\bar{BD}}^{2}}\\ & =\sqrt{4\delta^{2}-4\delta^{4}+4\delta^{4}}=2\delta \end{array}$$The half length of arc $\overset{⌢}{CD}$ is approximated to $\bar{BC}=2\delta$.

Next, we find the radius of the blue circle $⊙r_{o}$. As shown in Figure 3, point *E* is the middle of the segment $\bar{AB}$.

In Figure 3, ∠AOB=π−2β, $\bar{OA}=1$, and $∠Ar_{o}E=\frac{∠Ar_{o}B}{2}=\frac{J+1}{2}\alpha =\frac{(J+1)\pi }{2J+3}$. In △AOE and $△AEr_{o}$:(14)$$\begin{array}{cc} \bar{AE} & =\sin\frac{∠AOB}{2}=\sin\frac{\pi -2\beta }{2}=\cos\beta , \end{array}$$(15)$$\begin{array}{cc} \;\;\bar{Ar_{o}} & =\bar{AE}\csc∠Ar_{o}E=\cos\beta \;\csc(\frac{J+1}{2J+3}\pi ). \end{array}$$Thus, the number of iterations on the arc length $\overset{⌢}{BC}$ is
(16)$$\begin{array}{cc} \;\;\;\;\;\;J_{\delta } & =\left\lceil \frac{\overset{⌢}{BC}}{\bar{Ar_{o}}\cdot \alpha }\right\rceil ≃\left\lceil \frac{\bar{BC}}{\bar{Ar_{o}}\cdot \alpha }\right\rceil \\ \; & =\left\lceil \frac{(2J+3)\delta }{\pi }\sec\beta \sin(\frac{J+1}{2J+3}\pi )\right\rceil \end{array}$$
(17)$$\begin{array}{cc} \; & ≃\left\lceil \left(\frac{\csc\beta }{2}+\frac{4}{\pi }\right)\delta \right\rceil , \end{array}$$
where ⌈⌉ denotes ceil function. Therefore, after several iterations with a number in $\left[J+1-J_{\delta },J+1+J_{\delta }\right]$, the success rate in $\left[1-\delta^{2},1\right]$ will be achieved.

We show the relationship between 1/λ and the minimum number of queries for a given success rate in Figure 4, which means that if the desired rate of success is not high, the number of queries is correspondingly reduced.

## 4. Robust Quantum Search with Uncertain Number of Targets

Now, consider the situation where the ratio M/N is not known. Our goal is to find the marked state with high success. Here, we propose a robust search version of the Grover–Long algorithm. In our algorithm, the error is bounded by a parameter δ over $\Delta /\lambda_{0}$. In fact, our algorithm degenerates to the original Grover–Long algorithm if Δ=0. We proceed our algorithm as follows. The initial state is prepared to the superposition state |s〉. Then, we find out the target state |T〉 with success probability higher than $1-\delta^{2}$, in which the overlap $\langle s|T\rangle =\sqrt{\lambda }e^{i\xi }$ is nonzero (ξ is the phase difference between |s〉 and |T〉) and δ∈[0,1]. We provide the oracle operator $I_{\tau }$ which will flip the ancilla qubit if it matches the target state, that is, $I_{\tau }|T\rangle |a\rangle =|T\rangle |a⊕1\rangle$ for a=τ and $I_{\tau }|\bar{T}\rangle |a\rangle =|\bar{T}\rangle |a\rangle$ for a≠τ, while |T〉 are orthogonal to $|\bar{T}\rangle$. Next, we prove how to extract $|\bar{T}\rangle$ by querying $J+1-J_{D}$ times with the successful probability higher than $1-\delta^{2}$. This algorithm is shown as follows.

Suppose there is a database without exact λ but, rather, its upper and lower bounds are denoted as $\lambda_{0}\leq \lambda \leq \lambda_{0}+\Delta$. If we plug the lower bound $\lambda_{0}$ as the “overlap” into the Grover–Long algorithm, we will obtain
(18)$$\beta_{0}=\arcsin\left[\sqrt{\lambda_{0}}\right]\leq \arcsin\left[\sqrt{\lambda }\right]=\beta ,$$
then
(19)$$J=j_{op_{0}}=\left\lfloor \frac{\pi -2\beta_{0}}{4\beta_{0}}\right\rfloor \geq \left\lfloor \frac{\pi -2\beta }{4\beta }\right\rfloor =j_{op},$$
which obeys Equation (Equation 2), that $J\geq j_{op}$. Thus, J+1 can be chosen as the number of iterations in Grover–Long algorithm, but when plugging *J* and $\beta_{0}$ into Equation (Equation 3), we will not obtain the right matching phase. Thus, we will not obtain the right rotation angle α in Equation (Equation 9) for each iteration along the axis $\vec{l}$ in Equation (Equation 10). Instead, we will obtain
(20)$$\begin{array}{cc} \alpha_{0} & =4\arcsin\left[\sin\left(\frac{\phi_{0}}{2}\right)\sin\beta \right]\\ & =4\arcsin\left[\sin\left(\frac{\pi }{4J+6}\right)\sqrt{\frac{\lambda }{\lambda_{0}}}\right]\\ & \leq \frac{\pi }{2J+3}+\tan\left(\frac{\pi }{4J+6}\right)\frac{\Delta }{2\lambda_{0}}. \end{array}$$The angle difference dα between $\alpha_{0}$ and α is
(21)$$d\alpha =\alpha_{0}-\alpha \leq \tan\left(\frac{\pi }{4J+6}\right)\frac{\Delta }{2\lambda_{0}}.$$The total rotation angle difference is
(22)$$\begin{array}{cc} D\alpha & =(J+1)d\alpha \end{array}$$
(23)$$\begin{array}{cc} \; & \;\;\;\;\;\;\;\leq \left(J+1\right)\tan\left(\frac{\pi }{4J+6}\right)\frac{\Delta }{2\lambda_{0}}. \end{array}$$The angle Dα is the overcooked angle ∠CoB or arc $\overset{⌢}{CB}$, shown in Figure 2. We define this overcook as 2δ:(24)$$0\leq D\alpha \leq \left(J+1\right)\tan\left(\frac{\pi }{4J+6}\right)\frac{\Delta }{2\lambda_{0}}=2\delta .$$We then reduce the number of iterations in order to improve the success rate. The relation between the number of iterations and the error is given by Equation (Equation 16). Thus, the reduced number of iterations is
(25)$$\begin{array}{cc} J_{\delta } & =\left\lceil \left(\frac{\csc\beta }{2}+\frac{4}{\pi }\right)\delta \right\rceil \\ & \geq \left\lceil \left(\frac{\csc\left(\arcsin\sqrt{\lambda_{0}+\Delta }\right)}{2}+\frac{4}{\pi }\right)\delta \right\rceil \end{array}$$
(26)$$=\left\lceil \left(\frac{1}{2\sqrt{\lambda_{0}+\Delta }}+\frac{4}{\pi }\right)\delta \right\rceil =J_{D}$$
which drives the final state into the interval of arc $\overset{⌢}{C^{\prime }B}$ instead of $\overset{⌢}{CB}$, so the success rate of the final search is greater than $1-\delta^{2}$.

The flowchart of our Algorithm 1 is listed as follows:
**Algorithm 1** Robust quantum search with uncertain number of targets.
for a given database, just know the lower bound $\lambda_{0}$ and upper bound $\lambda_{0}+\Delta$ of λ=M/N, ($\lambda_{0}\leq \lambda \leq \lambda_{0}+\Delta$).
**Begin**
Calculate $\beta =\arcsin(\sqrt{\lambda_{0}})$Calculate $J=\left\lfloor [(\pi /2-\beta )/(2\beta ))]\right\rfloor$Calculate $\phi =2\arcsin\left(\frac{\sin\left(\frac{\pi }{4J+6}\right)}{\sin\beta }\right)$Calculate $\delta =\left(J+1\right)\tan\left(\frac{\pi }{4J+6}\right)\frac{\Delta }{4\lambda_{0}}$Calculate $J_{D}=\left\lceil \left(\frac{1}{2\sqrt{\lambda_{0}+\Delta }}+\frac{4}{\pi }\right)\delta \right\rceil$Obtain the search operator:$I_{\tau }=I+\left(e^{i\phi }-1\right)\left|\tau \rangle \langle \tau \right|$$I_{0}=I+\left(e^{i\phi }-1\right)\left|0\rangle \langle 0\right|$$Q=-HI_{0}HI_{\tau }$Implement the search operator *Q* on the initial state $|\psi_{i}\rangle$ for $J+1-J_{D}$ times.Make measurement of the final state and one will find out the marked state with the probability greater than $1-\delta^{2}$.
**End**


Compared with other algorithms, our algorithm maintains quadratic speedup. As shown in Figure 5, our algorithm and Yoder–Low–Chuang algorithm have the same query complexity $O\left(\sqrt{1/\lambda }\right)$ as the standard Grover’s algorithm. Under the condition of output success rate greater than $1-\delta^{2}=0.96$, our algorithm makes eight queries while the Yoder–Low–Chuang algorithm makes 10 queries and the π/3-algorithm makes 160 queries for $1/\lambda_{0}=100$. For $1/\lambda_{0}=4000$, our algorithm makes 46 queries while the Yoder–Low–Chuang algorithm makes 64 queries and the π/3-algorithm makes 6437 queries. For $1/\lambda_{0}=$ 10,000, our algorithm makes 72 queries while the Yoder–Low–Chuang algorithm makes 100 queries and the π/3-algorithm makes 16,094 queries. Both our algorithm and the Yoder–Low–Chuang algorithm have the same query complexity $O\left(\sqrt{1/\lambda }\right)$ as the standard Grover’s algorithm, while the query complexity of π/3-algorithm is scaled as 1/λ.

## 5. Discussions

In our algorithm, the infimum bound of the success rate is described by Equation (Equation 24). The infimum bound of the success rate is related to $\Delta /\lambda_{0}$. In Figure 6, we show the relationship between the lower success rate and the overlap rate at different uncertainty Δ. When the uncertainty is zero, 100% success rate will be achieved every time, and our algorithm degrades to the standard Grover–Long algorithm. The curve increases sharply with the increase of λ, and the success rate is above 80% when the uncertainty is double λ. When the uncertainty is the same order as λ, one can still achieve a high success rate, as high as 95%.

Therefore, our algorithm has a high tolerance to the uncertainty of the ratio λ=M/N. In order to see the performance of the robustness of our algorithm clearly, especially when the overlap λ is small, we provide Figure 7, which shows the relationship between probability and 1/λ. By taking $1/\lambda_{0}$ to infinity, one can see that when Δ equals zero, our algorithm degrades to the original Grover–Long algorithm. For Δ equals to $0.8\lambda_{0}$, the success rate of our algorithm exceeds 97%, and our algorithm has a success rate of more than 90% when Δ equals $1.6\lambda_{0}$, and a 78% success rate when Δ equals $2.4\lambda_{0}$. Even if Δ equals 3.2$\lambda_{0}$, our algorithm still has success rates above 60%.

In the worst-case scenario, one knows nothing about the rate M/N. Then, one has to run the quantum counting algorithm to estimate *M*, which will have an uncertainty $\sqrt{M}$. Our algorithm works well in this case, with a success rate above 96% and keeping the total $O\left(\sqrt{N}\right)$ query complexity as in Grover’s algorithms. After running the quantum counting algorithm, one obtains the ratio with uncertainty, that is $\left(M\pm \sqrt{M}\right)/N$. Thus, $\Delta =\lambda_{0}/\sqrt{M}\leq \lambda_{0}$. Plugging this value into Equation (Equation 24), the result shows that the success rate of this algorithm is higher than 96%.

Our algorithm can be used as a subroutine in any case where amplitude amplification [8] or Grover search are used [24,25,26,27].

In summary, we propose a robust quantum search algorithm with both the advantage of simple search operators, and high success rate over a wide range of M/N values. Therefore, it provides many potential applications [28,29,30,31,32] in future quantum computing.

## Figures and Tables

**Figure 1 entropy-23-01649-f001:**
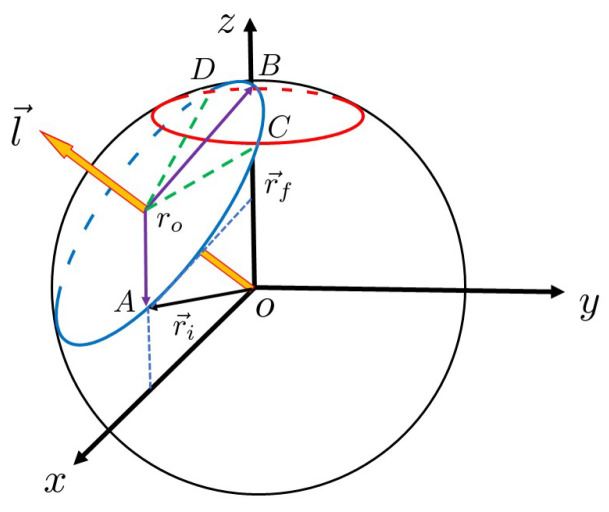
Geometrical description of Grover–Long searching algorithm. During each iteration, the state vector $\bar{OA}$ rotates around $\vec{l}$ with an angle α. After J+1 times iteration, $\bar{OA}$ overlaps with $\bar{OB}$, the target state |τ〉. In this picture, the probability of finding the marked state is $(z_{A}+1)/2$ [11], where $z_{A}$ is the *Z* component of point *A*.

**Figure 2 entropy-23-01649-f002:**
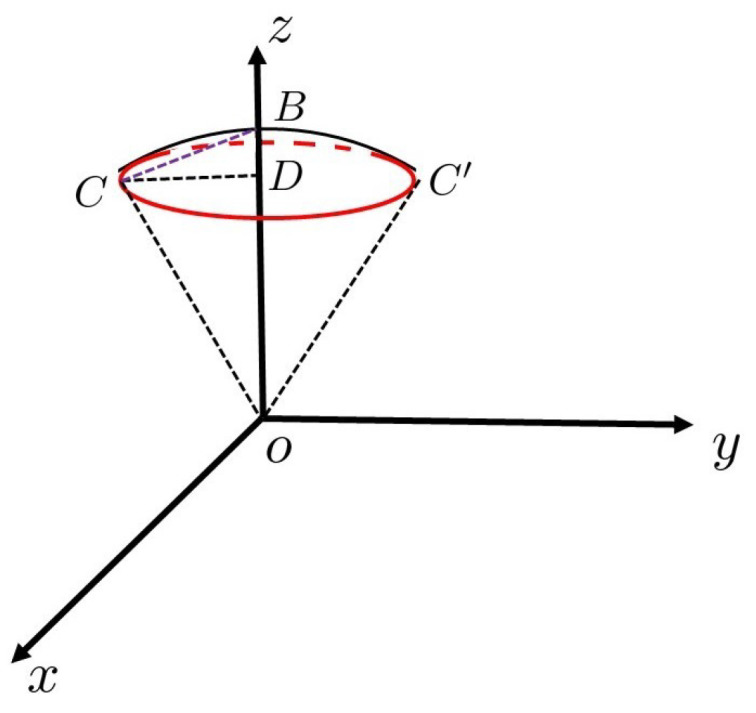
Error circle in a cone.

**Figure 3 entropy-23-01649-f003:**
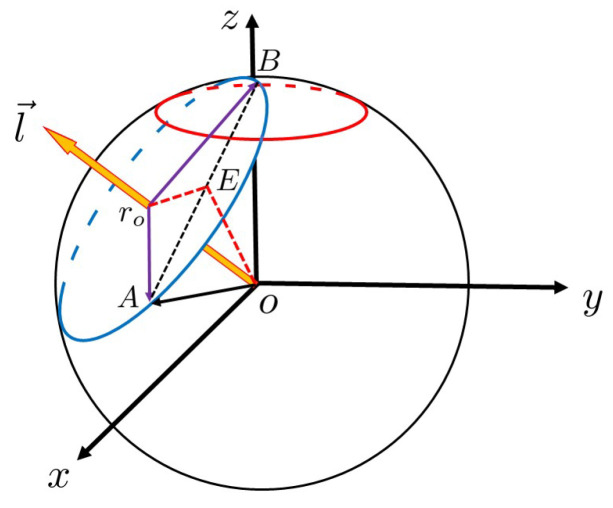
Geometrical description of quantum searching algorithm. Based on Figure 1, connect points *A* and *B* for segment $\bar{AB}$, then take the midpoint *E* of $\bar{AB}$, and connect points $r_{o}$, *E* and *E*, *o*.

**Figure 4 entropy-23-01649-f004:**
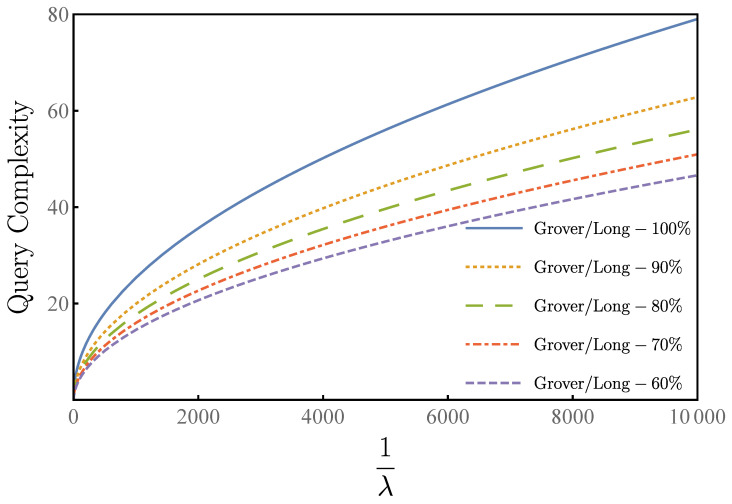
A comparison of Grover–Long algorithm with different success rates $1-\delta^{2}$. The query times versus the overlap $\lambda ={\left|\langle \tau |s\rangle \right|}^{2}$ of the target state |τ〉 with the initial state |s〉. Grover–Long-100% (blue) for the 100% success rate. Grover–Long-90% (orange) for $\delta^{2}=0.1$. Grover–Long-80% (green) for $\delta^{2}=0.2$. Grover–Long-70% (red) for $\delta^{2}=0.3$. Grover–Long-60% (purple) for $\delta^{2}=0.4$.

**Figure 5 entropy-23-01649-f005:**
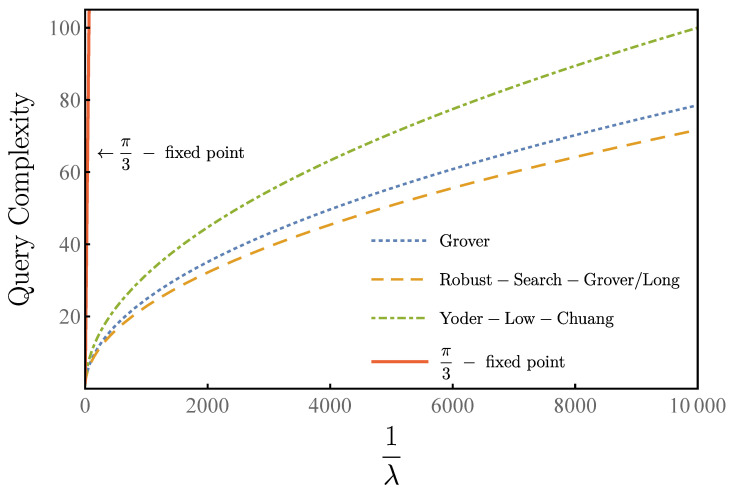
Query complexity versus 1/λ with $\delta^{2}=0.04$ for our algorithm (orange), Yoder–Low–Chuang algorithm (green), the π/3-fixed-point algorithm (red), and the origin Grover’s algorithm (blue).

**Figure 6 entropy-23-01649-f006:**
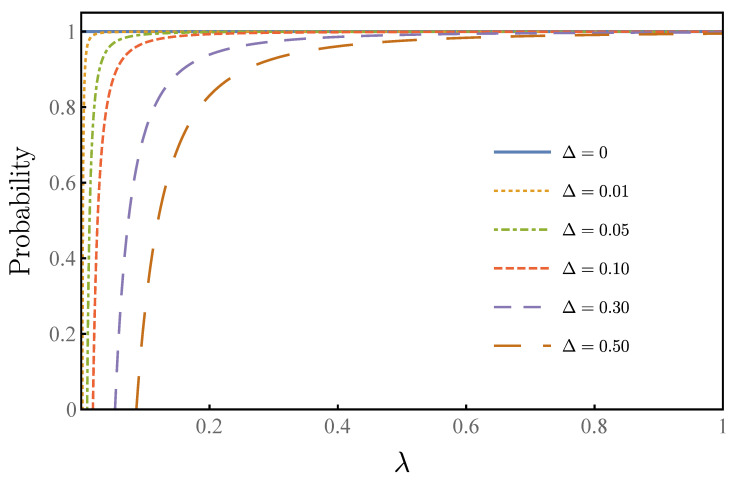
The horizontal coordinate of the figure is the proportion λ, and the ordinate is infimum bound of the probability of success. In the figure, we use different colors to mark different deviations of Δ. One can see that when the deviation is zero, the success rate is 100% each time, which corresponds to the standard Grover–Long algorithm. For the same proportion λ, the smaller the deviation, the higher the success rate.

**Figure 7 entropy-23-01649-f007:**
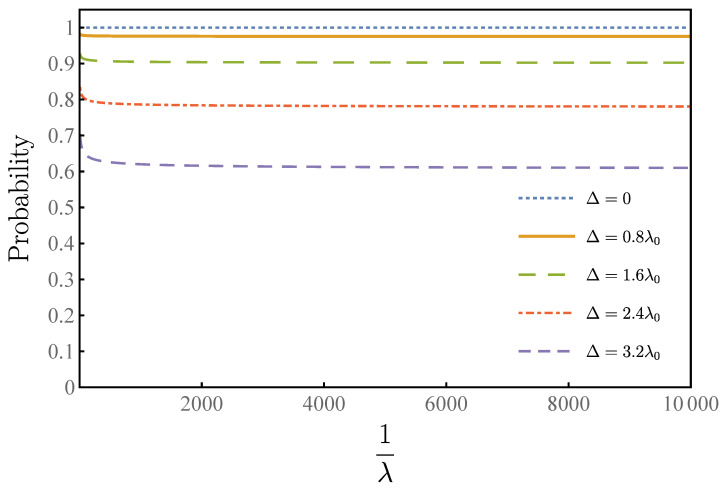
This figure shows that our algorithm has the robustness for the uncertainty of λ through the relationship between the deviation and the infimum bound of the optimal success rate, where horizontal axis is 1/λ and vertical axis is success rate, with different color curves representing different deviations.

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
