# Peer review of "Robust Quantum Search with Uncertain Number of Target States"

_entropy, 2021, doi:10.3390/e23121649_

Round 1

Reviewer 1 Report

The authors present a work which could be without any doubts interesting for the community. However before the publication in Entropy, the paper has to be improved. 

The English must to be improved (especially the “consecution temporum”). (note also that you didn't include the space before the citation number mostly everywhere)

The introduction is to much condensed and lacks to discuss the spatial search for multiple elements. For instance, in some natural systems, a spatial searching protocol has been recently proved which works in the multi-targets framework. In that case, it is not needed a priori the number of marked elements, still keeping the same advantage respect to the classical case (Phys. Rev. Lett. 124, 180501). Also Child and Portugal got some interesting results on this topic.

Section 2 must be largely improved. The authors have not properly introduced all the needed definitions. Just to make an example, \ket{\tau} is not defined. I would review ‘entirely’ (I really mean it) section 2 and would make it more accessible and pedagogical, and correct ( do not include definitions is an error). If someone is not familiar with quantum searching, would be not able to follow sec. 2 at all. 

Again the main body of the paper (sec 3-4) is to condensed and very difficult to follow, I was not entirely able to check the rightfulness of your results. In particular, at line 169 you write “Compared with other algorithms, our algorithm maintains quadratic speed up.“ Then you refer to your Fig 5. But I don't see any fit. Which are the prefactors? How do they change? Which fit did you use? Can you compare your numerical fits with an analytical proof?  

I could certainly be able to review again the paper after major revision. At this stage I am not able to seriously asses your work. 

Reviewer 2 Report

See attached

Reviewer 3 Report

The authors address the problem of provinding a variant of the Groover search algorithm in which the ratio between number of elements of the database and number of marked items is unknown. Only an interval is known. The authors analyze the performance of their algorithm for different values of the parameters characterizing the Interval.

The content of this manuscript can be of interest for readers of Entropy working in quantum information theory. Some minor comments below.

  • In page 6, line 122, there is an undefined phase in the probability amplitude between S and T.
  • In figure 6, the \Delta = 0 curve is difficult to visualize. Please, try to improve.
  • In Long ‘s paper (ref 11), M=1. The authors should explain in their review how things change when M is greater than 1.

Round 2

Reviewer 1 Report

The authors have sensibly improved the quality off the paper although they have could do more. By the way the paper seems to fit the journal requirements and then I suggest the publication.